# Generative Editing via Convolutional Obscuring (GECO): A Generative Adversarial Network for MRI de-artifacting

## Abstract

Machine learning models trained on imaging data have empirically shown an ability to detect complex and ill-defined artifacts, such as ones resulting from differences in which machines medical images were taken by, to a high degree of accuracy. Such artifacts are invisible to even the expert eye, yet readily lead machine learning systems to focus on spurious correlations rather than relevant features. Several forms of these spurious correlations have been reported to interfere with cross-institutional medical imaging research and model generalizability, representing one of the most significant open problems in medical machine learning for over a decade. Here, we present Generative Editing via Convolutional Obscuring (GECO), a Generative Adverserial Network for MRI deartifacting. With the application of GECO, classifiers' ability to detect these spurious correlations went from 97% down to a detection accuracy nearly equal to random chance. We also observe over 98% structural similarity between the original and deartifacted images, indicating the preservation of the vast majority of non-spurious information contained in the original images. In addition to solving the known problem of removing artifacts from MRI images, this method opens the door to potentially removing many other types of spurious correlations from images and other data modalities across many fields.

## 1 Introduction

In the field of medical imaging there is a strong desire to train AI models for disease and pathology detection. Magnetic Resonance Imaging (MRI) is a powerful, high-resolution modality which requires no radiation exposure yet still provides valuable clinical information to medical professionals, particularly with brain pathologies given the difficulty of imaging within the skull using X-rays. The complexity of medical scans and their information-richness have made them a key target of machine learning research. However, when machine learning models are trained on medical images (MRI or otherwise) they often encounter issues from spurious correlations. For example, it has been known for years that AI models easily memorize features of scans such as the particular model of machine used to take the scan. Because of this, models often fail to generalize beyond the training dataset (Badgeley et al., 2018). The specific artifacts used by models to predict machine type and manufacturer are generally poorly understood and often invisible to the naked eye, so removing them from images is a complex task. Here, we propose, GECO, a method for automatically removing said artifacts, by leveraging generative deep learning techniques to 'harmonize' an input image. We develop a generator network that is capable of modifying an input MRI image to prevent a classifier from being able to identify the machine manufacturer corresponding to medical images, while also not affecting the quality of the image.

GECO takes in a MRI image and outputs via a generative network a 'deartifacted' version of the original image. This deartifacted product retains virtually all of the structure of the initial scan, but eliminates or significantly reduces any identifying marks that would allow a deep learning model to determine the spurious correlation from the original image. Our approach for doing so is to use a GAN, which is trained to take raw MRI scans and remove as much as possible these spurious correlations. Note that this is not the same process as removing scanning artifacts such as zipper or herringbone artifacts that affect image quality, but rather the intent is to remove features that are

likely invisble to the human eye and impact a classifier's ability to categorize the image. Our results focus on image artefacts corresponding to different MRI manufacturers and field strengths. In each case, across the training of these GANs we reduce the classification accuracy from nearly perfect down to a level essentially equivalent to random guessing on the part of the classifier. We further demonstrate that GECO is similarly capable of "fooling" expert neuro-radiologists.

## 2 RELATED WORK

Examining related studies in this area, the work of Badgeley et al. (2018) serves as one of the primary inspirations for this project, and we set out to resolve some of the relevant issues via a state-of-the-art deep learning solution. Zech et al provided an overview of the difficulties in training classifiers on radiological data, and discussed potential reasons for the observed generalization gaps. Bagul et al. (2017) and Coram et al. (2016), achieved performance similar to radiologists on specified tasks. The latter case saw at least some successful generalization between institutions' patient populations - though it is not unsurprising that some cases would successfully generalize despite artifacts. Work in this field indicates that CNNs have the capacity to attain a near-expert level of performance in the absence of spurious correlations. Klein et al. (2021) developed an MRI image pre-processing pipeline capable of detecting metadata in a scan using purely the image data alone, and we use their architecture as a baseline for our classification task.

In the realm of MRI, different types of artifacts arise due to various reasons. Researchers have been developing sophisticated methods to improve the quality of MRI scans. For instance, Lim et al. (2023) devised an innovative approach using a GAN to mitigate motion artifacts. Motion artifacts are prevalent and can severely degrade the quality of the scans. In another study, Berrington et al. (2021) focused on the removal of spurious echo artifacts by employing a sensitivity encoding method. Their method efficiently estimated these artifacts and successfully removed them. Schramm & Ladefoged (2019) conducted a comprehensive review of multiple strategies that have been successful in correcting metal artifacts. Their survey highlighted techniques, which have been instrumental in improving image quality in the presence of metal.

Interestingly, despite the multitude of studies focusing on various artifact types, to the best of our knowledge, there is limited research on the removal of artifacts whose correlation properties and origins are too complex to cleanly define. Yet such artifacts are well-established to significantly impair the efficacy and generalizability of models trained on medical images. In medical imaging specifically, scanner-related artifacts can stem from calibration errors, hardware inconsistencies, or other scanner-specific issues, but they can also be produced by nuances of the software used to construct the images. There is not yet any known way to cleanly define general artifacts of this complexity, and in our view it is most likely the case that multifactorial processes produce the complex spurious correlations which we attack in this paper. It is pivotal to identify and address these artifacts to ensure the consistency and reliability of MRI scans across different imaging platforms, and by extension between institutions. By producing a method which can eliminate such subtle and complex yet impactful artifacts, we anticipate that GECO and future methods based on it will have broad applicability to a variety of other sources of spurious correlations.

In contrast with the work done by Yan et al. (2014) on photo editing via neural networks which performed generic edits, our architecture for image editing is a one-shot, fully convolutional process that has no reliance on segmentation maps or other learning tasks in order to generate the edited image. Zhu et al. (2016) also published a generative approach to photo editing, but this was again in the context of generic image editing and their method produces divergence between the input and generated images which are not desirable for de-artifacting. Pathak et al. (2016) saw some success at generating blocked out data from general images, though once again encountered the traditional difficulties observed with generative networks such as blurriness in high-detail areas.

In summary, related approaches to image editing required some level of manual defining of desired changes, which is highly problematic in the case of artifacts with unknown and complex distributions. Our work overcomes this difficulty by directly detecting the artifacts' distributions and editing them out all within a single, unsupervised architecture which we call Generative Editing via Convolutional Obscuring, or "GECO". In this paper, we demonstrate that GECO is a powerful approach for removing unwanted information from images with complex distributions.

## 3 METHODS

### 3.1 TECHNICAL APPROACH

Our process begins with a large collection of MRI images, whose metadata is known to us but not visible to humans in the images themselves. Two pieces of metadata were selected on the basis of their complex impact on the images themselves: scanner manufacturer and scanner field strength. MRI machines have designs which vary between manufacturers and play into the physics of the scanners, and additionally the strength of the magnetic field generated by a scanner significantly impacts the images produced. For classifying by manufacturer, on the basis of data availability we selected four companies: GE Medical Systems, Siemens, Philips Medical Systems, or Imaging Biometrics. These companies are either machine manufacturers or providers of processing proprietary software, or both. In this paper we use "manufacturer" to mean the label which is found in the "manufacturer" entry for an image's DICOM file (a standard format in radiology). For field strengths, most commercial models today operate at either 1.5 Tesla (T) or 3T, and so these were the natural classes for the network to operate with in the field strength experiments. It is important to observe that this is a classification problem rather than a real/fake detection problem as is more common in GAN training systems. We assessed only one of the two classification problems - manufacturer and field strength - in a given experiment, so separate results are provided for each.

The GECO generator that we found to be the most efficient is based upon the U-net architecture. Developed by Ronneberger et al. (2015b), U-nets have proven extremely effective for image segmentation and a variety of other problems. The overall structure of our GAN is inspired by Cycle-GAN (Efros et al., 2017). Our pixelwise loss has similarities to the identity loss term included in CycleGAN.

The generator network is given only the original MRI scan itself, and outputs a 'deartifacted' scan which retains as much as feasible the initial features of the image. With the generator, we use a Tanh function as the final activation to squeeze the output between -1 and 1. The classifier network follows the structure used by DeepDicomSort (Klein et al., 2021), employing a similar set of six convolutional layers, along with two dense layers on top for generating the final machine-type classification scores. Our core architecture is shown in Figure 4.

The structure of the losses shown in Figure 5 pit the generator and classifier networks against one another such that improving the loss of one worsens the loss of the other, with the goal that this teaches the generator network a strong representation of what 'deartifacted' images look like without having to provide examples ourselves. We define our learning target for this problem by developing a generator loss with two distinct components: the first is a sum of squared pixelwise errors between the input image and the output image, and the second is a cross-entropy style loss applied to the output of our classifier network. This 'inverse' cross-entropy loss is low when the classifier outputs a uniform probability distribution, and goes to infinity as any one class probability drops to zero (note the similarities to the TripleGAN structure (Li et al., 2022)). Note, as this portion of the loss is our own formulation, we provide a contour plot for the two-class case in Figure 5 to aid in conceptualization. The generator loss can be expressed as follows, where m is the total number of pixels in an image, c is the number of different classes, Gen.($\cdot$) is the output of the generator on a given input, Clas.($\cdot$) is the output of the classifier on a given input, and MSE is the mean-squared-error:

$$L_{gen.} = \frac{-1}{c} \sum_{i=1}^{c} log(Clas.(Gen.(X))) \\ + \beta \sum_{i=1}^{m} MSE(X_i, Gen.(X)_i). \tag{1}$$

Note the beta term associated with the MSE component of the loss, which we use as a hyperparameter to tune the importance of the generator's two tasks. We interpret this loss as being split into a classification and a discrimination component, replacing the discriminator network from Triple-GAN (Li et al., 2022) with discriminatory loss in our case for simplicity and stability. The classifier network is only used during training and evaluation time, and takes in as input a MRI scan from

the generator. The classifier outputs a predicted class label which is compared to the true label to generate the cross-entropy loss that is used to update the classifier. The loss that guides the classifier training process is calculated as follows:

$$L_{disc.} = CE(Y, Clas.(Gen.(X))) \tag{2}$$

The final product of this process is the trained generator network, which should be able to deartifact an MRI scan given only the raw unpadded image, as all the learnable weights are in fully convolutional layers that do not care about the pixel size of the image.

In this work we considered the following architecture variants: a core generator/classifier pair, a max-pooling classifier, a three-player architecture, a residual generator and a multi-residual block generator. Further we explore four variants in addition to our core architecture.

### 3.1.1 VARIANT CLASSIFICATION ARCHITECTURES

In terms of the classifying part of our architecture, we explore two other variants. The first one includes a max-pooling layer after all convolutional layers, and for the other one we introduced a discriminator network to explore three-player architectures. For the max-pool classifier, the pooling layer guarantees that the dense head of the classifier will always receive N inputs where N is the channel size of the final convolutional layer. This allows the classifier to be scale invariant, so long as the features of the image are not vastly larger or smaller than the images on which the classifier is trained.

In the case of the three-player architecture, we explore using a learned discriminator network in place of the pixelwise MSE loss, resulting in a GAN structure with three distinct networks. In this structure the generator modifies the input images, and the generator output is fed to both a discriminator network that attempts to guess whether or not the image is unmodified or output from the generator and a classifier network that attempts to guess which type of machine the image was taken from. The discriminator is also fed real images from the dataset as a separate phase of training in each iteration. As a result, the discriminator and classifier are evaluated using cross entropy, and the generator is evaluated using a combination of our own loss function on the classifier's output, and the reverse of the binary cross-entropy loss on the discriminator's output. That is our generator loss becomes (where BCE stands for binary cross-entropy)

$$L_{trip.gen.} = \tilde{L}_{gen.} + BCE(Clas.(Gen.(X)), \vec{0}) \tag{3}$$

(where $\tilde{L}_{gen.}$ is used to tackle a variant on our initial problem). Thus, the generator attempts to both fool the classifier into being unable to predict the class of the output image, while also producing 'realistic' images as evaluated by the discriminator. A diagram of our variant three-player structure is shown in Figure 6.

### 3.1.2 VARIANT GENERATOR ARCHITECTURES

In combination with our classification architectures, we also explore variants on the architecture of our generator. The primary technique we explore here is the concept of a residual generator, that is learning the edits to make to an image instead of the raw image itself. To do this, we construct the generator as normal with all convolutional layers, but after generating the final output we scale it down by a learnable parameter $\alpha$ and add it to the initial input. Mathematically, we represent the output $X'$ of the generator on an image X as:

$$X' = X + \alpha * Gen.(X) \tag{4}$$

With this structure, we allow the generator to decide the magnitude of the changes required to fool the classifier, while not requiring it to learn the identity mapping that optimizes the MSE component of the loss. We also explore a multi-residual block structure, with three residual blocks of two convolutional layers each. We believe that it is worth exploring these options due to the naturally

residual nature of the problem the generator is being asked to solve–that is, applying minor edits to the image while maintaining most of the structure.

The generator architecture that proved most effective was a modified U-net (Ronneberger et al., 2015a). While in recent years U-nets have seen widespread use for image segmentation problems (Ronneberger et al., 2015a; Milletari et al., 2016; Litjens et al., 2017; Isensee et al., 2018; Oktay et al., 2018; Zhou et al., 2018), their use in this paper is a novel application.

## 3.2 BASELINE METHOD

There are no examples we were able to find in literature of relevant generative 'editing' networks that we are able to use a baseline comparison. Instead, we use a classification network as our baseline for performance on this task and compare our post-learning classification results to show the effectiveness of our generative network. To that end, we use the DeepDicomSort (Klein et al., 2021) architecture and apply transfer learning to replacing the dense head with a set of weights trained to classify the manufacturer of the machine used to take the scan instead of the type of scan being taken. We use the weights hosted on the project's github page, slightly adjusting the architecture to fit our data. We then train the model for three epochs on the padded 512x512 unmodified MRI scans, freezing all weights in the convolutional backbone. As an output, we return the predicted class label of the input MRI scan and train with a cross-entropy loss. We obtain a loss of 0.0325, and a test accuracy of 0.9884, showing that even with a short duration of training, it is possible to achieve near-perfect classification on this task and to place greater context on the final results of our generator.

## 3.3 DATASET

We obtained images in the DICOM format as well as manufacturer information from the Cancer Imaging Archive (TCIA). We use the Brain Tumor Progression and TCGA-LGG datasets. Our preprocessing steps are inspired by DeepDicomSort's (Klein et al., 2021) handling of the same dataset. The brain images might have different orientation so we adjust output direction and origin. Particularly, we focus on an Axial orientation. After constructing the whole 3D brain from a set of DICOM files, we slice the brain horizontally into NIfTI slices (Figure 1e). We only keep meaningful images that have fewer than 90% of values set to pure black or pure white, as a heuristic to remove slices of the very top or very bottom of the brain. Before training, we zero pad all images to have the size as the largest image of the set (which in our case is 512x512). The final dataset contains 21987 slices and consists of four classes: GE Medical Systems, Siemens, Philips Medical Systems, or Imaging Biometrics. The class distributions are relatively balanced (ranging from 22% to 28%), and we follow a 70/20/10 rule to split the training/validation/testing sets.

## 3.4 HYPERPARAMETERS

For our hyperparameter tuning process, we selected three variables to explore: generator learning rate, classifier learning rate, and beta (our tuning parameter for the two-component generator loss). We started our sweep for the classifier and generator around $5 \times 10^{-3}$ as this was near the value used in the DeepDicomSort classifier and it had empirically shown strong results. We tested log space sweeps of approximately seven values per parameter, and achieved the best performance with the combined parameter set of $5 \times 10^{-4}$ for the learning rate of the generator and the classifier, and a beta value of 0.02.

## 3.5 METRICS OF GECO PERFORMANCE

We evaluate GECO using three performance metrics: Classification Accuracy, Structural Similarity Index Measure (SSIM), and Peak Signal-to-Noise Ratio (PSNR). The first metric quantifies the ability of the generator to deartifact an image in terms of how often it is capable of fooling a classifier trained on the output distribution, while the second and third metrics quantify the ability of the generator to maintain the integrity of the input image throughout the de-artifacting process. Classification accuracy is calculated via running the generator on our test set, and feeding the generated images into the corresponding classifier to achieve class predictions. These predictions are then compared against the true labels of the images fed into the generator, and we return the simple ratio

of correct guesses to total images. Since we have four classes, the 'optimal' accuracy we expect is 0.25, representing effectively random guessing.

SSIM is calculated by computing similarity between an image before and after it is processed by the generator, with 1 being the maximum. Finally, PSNR is calculated by comparing the square of the maximum value of the input image to the mean-squared error between the input image and the generated image. This metric is generally reported in terms of 'decibels' on a log scale, with a potentially infinite PSNR if the two images are identical.

# 4 RESULTS

## 4.1 ASSESSMENT AND COMPARISON OF GAN STRUCTURES

We explored several structures for the GECO network based on a review of the techniques used for generative modelling and methods for improving accuracy and robustness. Our results in Table 1 on the core architecture show high SSIM and PSNR reflecting visual clarity in the generated images as seen in Figure 1, and an accuracy of 26% from the classifier showing a large dip in performance compared to the pretrained model. We note that in failed experiments such as hyperparameter choices which led to mode collapse, poor visual fidelity, or restrained editing, the final classifiers retained approximately 96% accuracy. This indicates that the quantiative results showing GECO's efficacy do not reflect a lack of opportunity for the model to learn to classify these images. If the generator fails to train successfully for one of the defined and detectable issues listed above, the classification accuracy remains at the baseline of approximately 96%, but when the generator trains successfully the classifier's accuracy is reduced significantly.

Next, we tested two residual networks, both the multi-residual and full residual, and we observe similar results to the core generator but with moderately degraded performance in terms of image quality and deartifacting efficacy Table (1). In addition to the residual generators, we tested a three-player architecture. This architecture frequently suffered from a mode collapse and instability during training. With the three-player architecture, we observed a slight improvement moving to two iterations of the classifier instead of one per generator iteration, however in the process we observed that the accuracy of the classifier during training did not increase as much compared to a one for one training regime. For comparison, the two iteration classifier achieved a maximum accuracy of 42% during training, while the one iteration classifier achieved a maximum accuracy of 47% during training. We also tested using three iterations for the classifier, but the SSIM decayed beneath the benchmark of 0.95 where image quality began to degrade.

From these results and the data in Table 1, we select the core architecture as our optimal structure.

## 4.2 EVALUATION OF GECO'S EFFICACY IN HAMPERING CLASSIFIERS

Next, we focus on analyzing the performance of the core architecture across both manufacturer-specific and field strength-specific experiments. In Figure 1 we show the SSIM and accuracy while

Table 1: Metrics on various successful architectures, with the best performance in each metric highlighted in bold. SSIM and PSNR correspond to preservation of image fidelity during the generation of the edited image, and Acc. corresponds to the accuracy of a separate classifier in identifying which of four classes an edited image belongs to.

| Experiment | Metrics | | |
|:---:|:---:|:---:|:---:|
| | SSIM | PSNR | Acc. |
| Core GAN | **0.986** | **37.75** | **0.256** |
| Residual GAN | 0.918 | 18.71 | 0.442 |
| Multi-Residual GAN | 0.933 | 19.93 | 0.514 |
| Three Player GAN (1 iter) | 0.982 | 32.58 | 0.396 |
| Three Player GAN (2 iter) | 0.981 | 33.27 | 0.378 |
| Three Player GAN, no MSE | 0.756 | 17.09 | 0.552 |

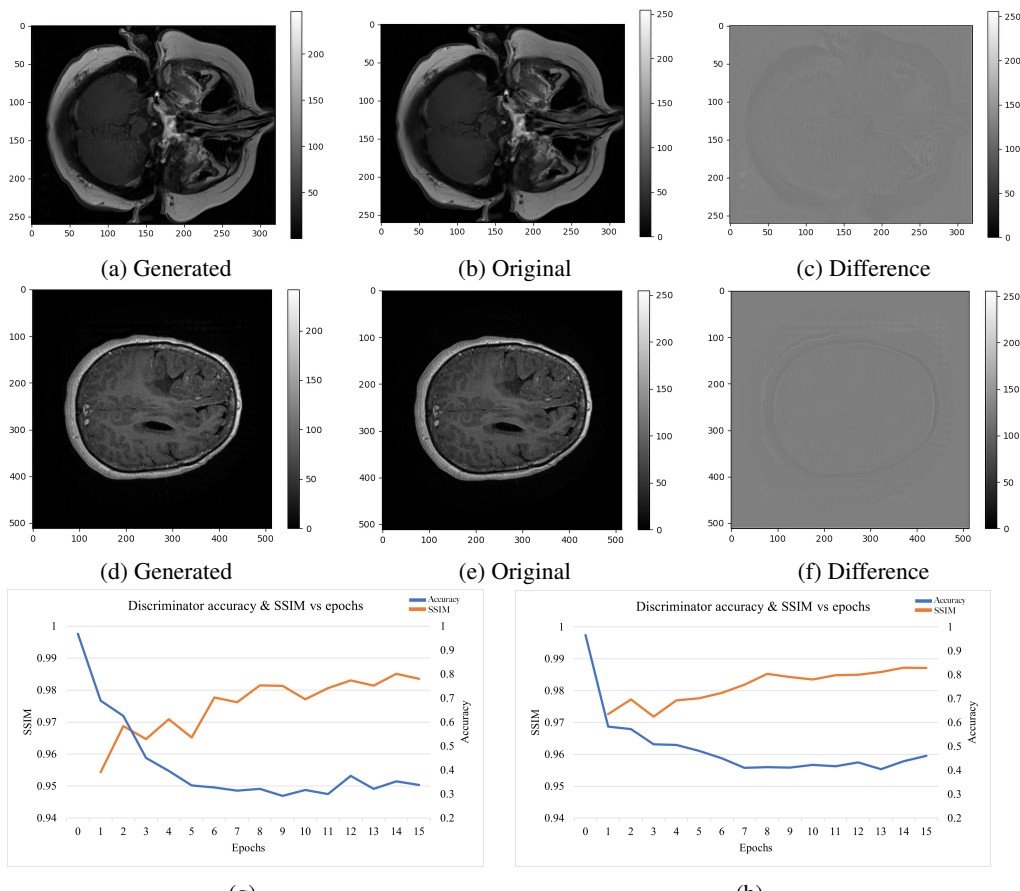

Figure 1: Visualization of two MRI scan slices being passed through GECO, and plots of performance metrics over the course of training GECO. (a), (b), and (c): Results of training a generator to eliminate the difference between images of different manufacturers (top) and field strengths (middle). Here (a) is the original image, (b) is the edited image taken from the GECO generator after 15 epochs of training, and (c) is a grayscale map of the pixel-wise differences between the original and generated (edited) image. (d), (e), and (f): The same progression as (a) through (c), but with a different input image as an additional example. (g) and (h) Change of accuracy and SSIM over epochs of training. Orange is the SSIM and is shown on the left vertical axis of each figure, and blue is the classifier accuracy shown on the right-hand vertical axis in each. Metrics are averaged over 5 independent training runs. In (g) the generator is trained to eliminate the difference between four manufacturer classes. The best metric values achieved when evaluating the 5 trained GECO networks on a test dataset were 0.986 for SSIM and 0.256 for accuracy after epoch 15. Averaged over the final 5 epochs the metric values are 0.337 for the accuracy and 0.983 for the SSIM. In (h) the generator is trained to eliminate the difference between field strengths (1.5T and 3T). Averaged over final 5 epochs the metric values are 0.431 for the accuracy and 0.986 for the SSIM. In the case of field strength this was essentially equivalent to the values for the individual training runs.

training the model for both field strength and manufacturer, showing an increase in SSIM and a decrease in accuracy essentially equalling a random classifier for manufacturer (Figure 1g) and field strength (Figure 1h). In Figure 2, we observe that the distribution of predicted labels becomes normalized across the true labels for both field strength and manufacturer. Finally, Table 2 provides indication of the ability of this model to perform well across several modalities of imaging.

We also explore methods to confirm the quality of our results and the behaviors of our models. We plot visualizations of the edits made by our generator network in terms of the difference between the input image and the generated image. We observe that these edits are in general very minor adjustments as we expect, and the effect appears to be most prominent around the skull. Using

Table 2: Metrics obtained on the manufacturer artifacts removal task reported by modalities. The number of samples from each manufacturer are given for reference.

| Modality | Accuracy | SSIM | Number of images | | | |
|---|---|---|---|---|---|---|
| | | | Siemens | Philips | Imaging Biometrics | GE Medical Systems |
| FLAIR | 0.101 | 0.983 | 145 | 39 | 120 | 143 |
| OTHER | 0.210 | 0.986 | 243 | 359 | 333 | 453 |
| T1 POST | 0.351 | 0.990 | 83 | 49 | 0 | 22 |
| T1 PRE | 0.347 | 0.984 | 71 | 71 | 232 | 47 |
| T2 | 0.305 | 0.988 | 186 | 85 | 112 | 132 |
| T2 FLAIR | 0.292 | 0.981 | 2 | 22 | 0 | 0 |

Figure 1f as an example, we also observe minor adjustments within the brain itself following along the edges of the image, but the strongest effects are in regions that are unlikely to change the medical information of the image.

### 4.3 EXPERT EVALUATION OF GECO PERFORMANCE BY RADIOLOGISTS

During the training process, we observed that when SSIM dips below approximately 0.95 we begin to see irregularities in the generated image that are clear to the naked eye. Conversely for SSIM values at or above that estimated threshold the two images appear virtually indistinct. While this level of analysis was suitable for the iterative process of model experimentation and construction, the final say in the success of our model's ability to maintain the semantically meaningful portions of a scan must be provided by trained radiologists. Thus, we collected data via survey asking experts to determine which scan they preferred between the unmodified and GECO-processed images as well as asking five neuro-radiologists to compare and score the visual quality of both types of images. We show our results in Figure 3.

From Figure 3a we observe that the large majority of cases show no significant preference between the provided images, and that in the cases where a preference is exhibited it is largely random which of the two images is preferred. Note that we refer to the original image as 'right' and the generated image as 'wrong' as we assume that our generator has not actually improved the image quality in any meaningful way, since it was not trained on any such task during this process.

This data is further backed up by Figure 3b, from which we observe that when asked to rate the quality of MRI scans on a scale of 1-10 our generated images achieved very nearly the same scores on average as unprocessed images. There is an overall very minor trend of generated images being perceived as slightly lower quality, but aside from radiologist 3 the scale of the effect is negligible. Thus, we conclude that our model has succeeded at its task of removing identifying artifacts without significantly impacting the image's value as a diagnostic tool.

## 5 CONCLUSIONS

In this work, we show that GECO is able to de-artifact MRI brain images by achieving a drop in classifier accuracy from ∼97% to ∼26% on the task of MRI manufacturer identification while maintaining the visual quality of images as measured by SSIM and PSNR. Similarly, for MRI field strength identification, GECO took the accuracy of a classifier from 97% to approximately 50% - and for that problem this was achieved in only a few epochs of training. This faster learning likely due to the fact that solving a two-class problem requires less data than a four-class problem in the case of removing manufacturer-specific artifacts. The magnitude of these artifact detection accuracy drops, indicates the generator network's strong efficacy in removing artifacts from scans, and in fact the final result is nearly equivalent to a classifier that is randomly guessing. Our architecture is particularly exciting given this level of efficacy was achieved with a relatively small number of convolutional layers in the model.

Applying GECO to similar problems of complex artifacts in CT and X-ray images are clear directions for follow-up research, and a more ambitious extension would be treating certain subtle physiological features as "artifacts". For example, there may be cases where researchers would like

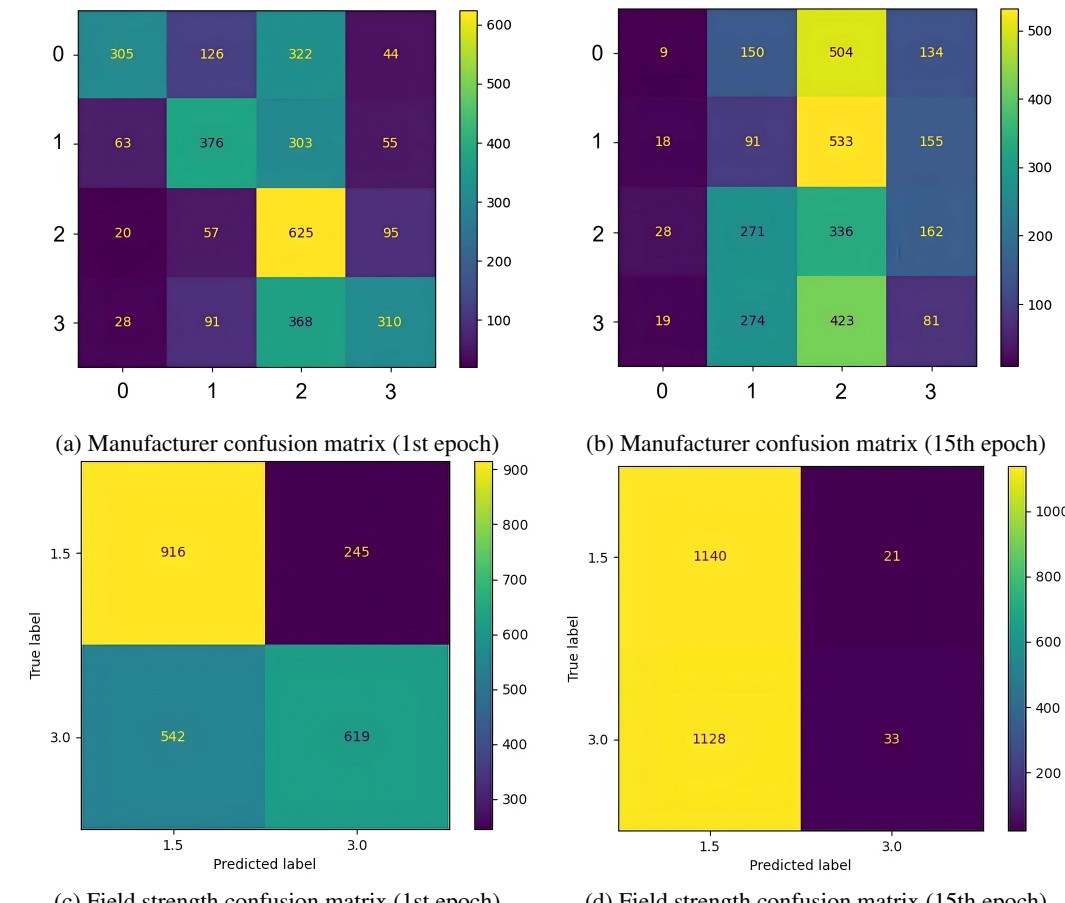

Figure 2: Confusion matrices demonstrating machine learning classifiers cannot effectively distinguish between original and GECO-edited generated MRI images. (a) and (b) Manufacturer confusion matrices, respectively for epochs 1 and 15. (c) and (d) Field strength classifier confusion matrices, respectively for epochs 1 and 15. For both cases, we observe that after training each true label has a similar distribution of predicted labels. The classes above for the manufacturers are: 0 – GE Medical Systems, 1 – Imaging Biometrics, 2 – Philips, 3 - Siemens, all corresponding to how they appear in the DICOM files.

to study one physiological or clinical feature, but a separate gene representing a spurious correlation produces subtle variation in brain architecture which are not visible to the naked eye yet still confound the efficacy of computer vision methods. In such a situation it may be possible to use GECO or a variant thereof to reduce the amount of information a machine learning system could possibly extract from those spurious correlation of variants of the gene, enabling the ML methods to more effectively learn the systems or processes which researchers and clinicians are interested in. Similar applications could be found in studying patients where separate disorders correlate with the feature(s) being studied, such as early-stage diabetic damage changing the appearance and physiology of organs subtly. In short, minimizing the extent to which machine learning methods can "shortcut" past the actual features of interest is likely to have broad applications in medical imaging.

Though here we have demonstrated GECO's ability to clear spurious correlations from medical imaging data for computer vision applications, there is great potential for applications of this method to solve similar problems with other data types. We expect that a broader range of potential applications may exist beyond medical imaging. This approach of generatively filtering spurious information, or future approaches based upon it, could not only be used for computer vision applications in various fields, but also in natural language processing. For example, the work of Koppel & Winter (2013) and Rocha et al. (2017) have demonstrated that NLP systems can accurately match individ-

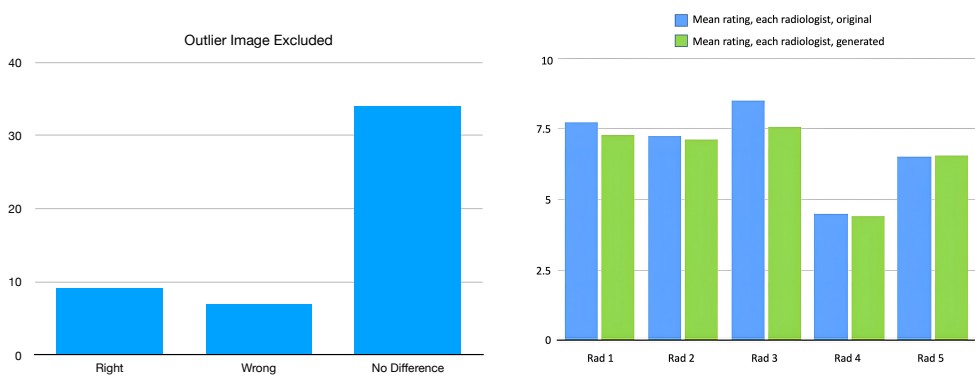

(a) Survey result: comparing paired MRI images

(b) Survey result: average scores of MRI image quality

Figure 3: Plots of results from surveying experts which demonstrate they are unable to effectively distinguish between original and generated images. (a) Summary of survey results when a group of 5 radiologists were asked to compare paired MRI images and the results of those same images being fed through a GECO trained to remove manufacturer artefacts. The radiologists could select from "Left", "Right", and "No Difference". Because image pairs were presented with the original and generated in random order with respect to left and right, we have translated the labels into "Right" and "Wrong" to show the results over 10 pairs. (b) Summary of additional results from the same survey, now showing differences in average scores of MRI image quality, in a separate section of the survey with half original and half GECO-edited images. Radiologists scored image quality on a scale from 1-10 with 1 the worst and 10 the best. Results are shown radiologist by radiologist, to account for variances in the baselines for different survey respondents. For example, Radiologist 4 ("Rad 4")'s average score for both original and edited images is less than 2/3 that of all others.

uals to even short pieces of text. With a generator modified for text data, a modification to GECO could potentially obscure the identities of authors. Further applications could potentially include difference-in-differences analyses (among other approaches) for causal inference in clinical trials and social sciences by aiding in accounting for externalities with complex and/or unknown distributions - especially in data like satellite or medical images.

To summarize, the approach at the core of GECO has potential for a setting where there is benefit in the precisely targeted elimination of specific correlates whose labels are known, yet whose distributions are not. It's strength lies in achieving high efficacy without excessive complexity. The GECO system's sole objective is to pass on as much data to downstream systems as possible while 'erasing' information which developers and users of other ML systems would like ignored. This 'restricts' downstream systems from detecting spurious correlations, and thus 'frees' them to learn the true distributions of interest.

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
