# A  APPENDIX

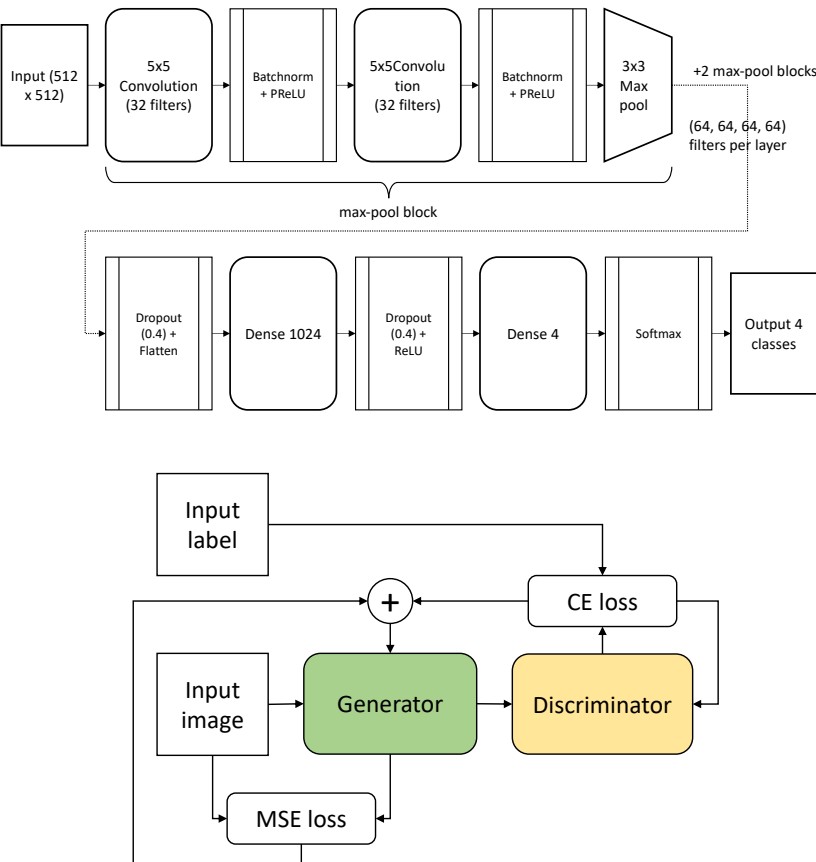

Figure 4: Extended Data Figure 1: Model Architecture. Top: layerwise architecture of our classifier, architecture initially from DeepDicomSort. Bottom: Depiction of the general structure for our GAN architecture.

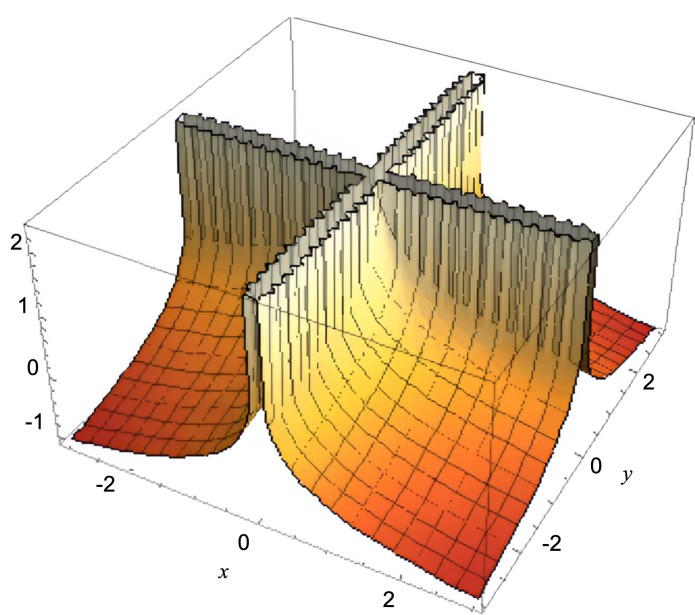

Figure 5: Extended Data Figure 2: Two-class example of our inverse cross-entropy loss structure.

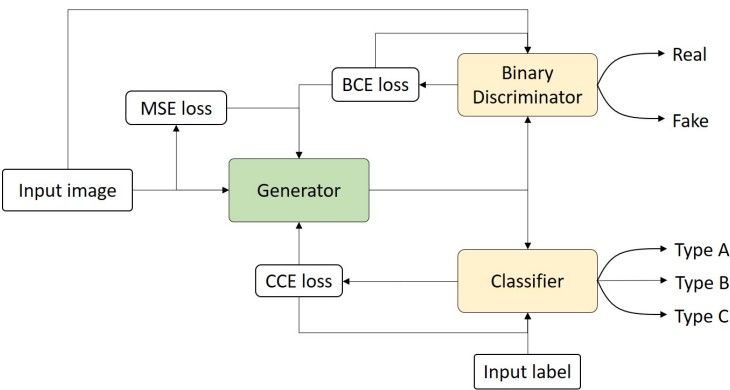

Figure 6: Extended Data Figure 3: Three-player architecture schematic