# OpenReview forum: "Generative Editing via Convolutional Obscuring (GECO): A Generative Adversarial Network for MRI de-artifacting"
_ICLR.cc/2025/Conference — Submitted to ICLR 2025_

### Official Review · Reviewer_WoRt · 2024-10-28

**Soundness:** 2
**Presentation:** 2
**Contribution:** 3
**Rating:** 3
**Confidence:** 4

**Summary:**

It is an urgent issue that machine learning systems tend to focus on spurious information in images while overlooking relevant features, as these spurious correlations hinder cross-institutional medical imaging research and model generalizability. The authors propose Generative Editing via Convolution Obscuring (GECO), a generative adversarial network designed to remove artifacts in MRI. By applying GECO, the classifier's ability to detect these spurious correlations dropped from 97% to near random-chance accuracy. Additionally, a high structural similarity was observed between the original and artifact-free images, indicating that the vast majority of non-spurious information in the original images was retained. This is an interesting study, however both the experiments and the writing lack sufficient rigor, with several issues present.

**Strengths:**

Removing spurious correlations from images is crucial for enhancing the generalizability of downstream tasks (such as classification, segmentation, and detection) and represents a unique and underexplored research direction. This study analyzes the impact across different classifier architectures and generative network structures, preventing classifiers from accurately capturing spurious information related to manufacturer and field strength categories.

**Weaknesses:**

1. The writing lacks rigor and needs improvement, including missing symbol explanations for X and Y in Equations (1)-(4). Additionally, there are misstatement errors, such as in line 423 where "achieved" should be replaced with "dropped."
2. The study lacks experimental comparisons with state-of-the-art methods and verification of spurious information removal for downstream tasks.
3. In Figure 1, the visualized “Differences” appear primarily as high-frequency edge information variations and background artifact removal. The edge high-frequency differences are largely due to the network's smoothing effect on outputs, which is common in generative tasks and does not necessarily represent spurious information. Removing background artifacts as spurious information has limited practical value, as downstream tasks typically apply background removal preprocessing to prevent background artifacts from impacting performance.

**Questions:**

1. These results are currently insufficient to demonstrate that this method effectively removes spurious correlations. Is there a better explanation for this? I believe background artifacts may be a significant influencing factor and should be excluded.
2. Also, why are there no comparative experiments with state-of-the-art methods in the classification stage or validation on downstream tasks?

---

### Official Review · Reviewer_wnU4 · 2024-10-28

**Soundness:** 2
**Presentation:** 3
**Contribution:** 2
**Rating:** 3
**Confidence:** 5

**Summary:**

The authors proposes a GAN approach to remove image artifacts arising from particular machine to take the scan.  The authors then present an ablation study from the Brain Tumor Progression and TCGA-LGG datasets and show the difference between generated images and original images are negligible.

**Strengths:**

The authors address an important that is important to the MRI community and in the real world.  The qualitative improvements are excellent and radiologists could not distinguish the results.

**Weaknesses:**

The authors have not addressed prior work that is related to this topic.

The task of removing cross-protocol and cross-scanner image attributes in scanners is known as 'Data Harmonization' and is well-studied in the MRI community.  Please see the following recent reviews:

https://www.nature.com/articles/s41597-024-02956-3

https://pmc.ncbi.nlm.nih.gov/articles/PMC10135601/

https://pubmed.ncbi.nlm.nih.gov/37084926/

There are many baselines the authors could have considered.

The authors could have also considered the field of domain adaptation/transfer for inspiration and baselines.

**Questions:**

Suggestions:
Use ` to start quotation marks in latex.

---

### Official Review · Reviewer_VvjK · 2024-11-03

**Soundness:** 3
**Presentation:** 2
**Contribution:** 2
**Rating:** 3
**Confidence:** 5

**Summary:**

The paper proposes GECO, which is based on Generative Adversarial Network for MRI deartifacting. Brain MRI images from 4 different manufactures are involved as the testbed. Multiple network architectures are explored and the harmonization seems to be successful as the classifier accuracy drops to around 25%.

**Strengths:**

1. The motivation of the paper is reasonable and removing artifacts does play an important role in medical imaging application.

2. The deartifacting effect of the GECO is expected, especially according to the dropping accuracy to around 25%, which is close to random guessing.

**Weaknesses:**

1. Structure of the proposed GECO lacks novelty compared to GAN, which involves one generator and one discriminator. Recently, diffusion models have been proposed and also applied in medical image translation and harmonization tasks [1].

2. It is said in the paper no previous baseline to compare. However, in my opinion, deartifacting task falls in medical imaging harmonization, where there are previous works [2][3]. It would be better for authors to conduct downstream tasks, for example, classification or segmentation, of the images harmonized using GECO, and the previous methods.

3. In related work (line 85-86), there is no evidence/papers supporting the sentence “artifacts are well-established to significantly impair the efficacy and generalizability of models trained on medical images”. It would be better to be more specific of how artifacts can affect the downstream tasks and include one or two examples.

[1] Özbey, Muzaffer, et al. "Unsupervised medical image translation with adversarial diffusion models." IEEE Transactions on Medical Imaging (2023).

[2] Bashyam, Vishnu M., et al. "Deep generative medical image harmonization for improving cross‐site generalization in deep learning predictors." Journal of Magnetic Resonance Imaging 55.3 (2022): 908-916.

[3] Abbasi, Soolmaz, et al. "Deep learning for the harmonization of structural MRI scans: a survey." BioMedical Engineering OnLine 23.1 (2024): 90.

**Questions:**

Artifacts in medical imaging can have multiple types. Some of them can significantly harm the generalizability of the models, for example, the text labels in the medical imaging. Based on the figure 1, the difference between the generated and the original images is minor. There is a concern that such artifacts would affect the performance on downstream tasks and to what extent. For example, one way to verify the harmonization process is to train the models using original and harmonized images, and test on set from another manufactures. Adding evaluation on potential downstream tasks can add more strengths to the paper.

---

### Meta-Review · Area_Chair_NMgZ · 2024-12-19

**Metareview:**

Reviewers are unanimous in their ratings, and there has been no response from authors.

All reviewers found that the submission is missing state of the art baselines, and appears to be unaware of the harmonization literature. It seems pertinent to either rebut these claims if possible, or provide comparisons to those methods.

**Additional Comments On Reviewer Discussion:**

Rebuttal of the reviewer claims would have been acceptable if they were incorrect.

---

### Decision · Program_Chairs · 2025-01-22

Reject